# A tunable refractive index matching medium for live imaging cells, tissues and model organisms

Tobias Boothe[1,2,3], Lennart Hilbert[1,2,3], Michael Heide[1], Lea Berninger[1], Wieland B Huttner[1], Vasily Zaburdaev[2,3], Nadine L Vastenhouw[1], Eugene W Myers[1,3], David N Drechsel[1†], Jochen C Rink[1*]

[1]Max Planck Institute of Molecular Cell Biology and Genetics, Dresden, Germany; [2]Max Planck Institute for the Physics of Complex Systems, Dresden, Germany; [3]Center for Systems Biology Dresden, Dresden, Germany

**Abstract** In light microscopy, refractive index mismatches between media and sample cause spherical aberrations that often limit penetration depth and resolution. Optical clearing techniques can alleviate these mismatches, but they are so far limited to fixed samples. We present Iodixanol as a non-toxic medium supplement that allows refractive index matching in live specimens and thus substantially improves image quality in live-imaged primary cell cultures, planarians, zebrafish and human cerebral organoids.

*For correspondence: rink@mpi-cbg.de

Present address: †Research Institute of Molecular Pathology, Vienna, Austria

Competing interests: The authors declare that no competing interests exist.

## Introduction

Live imaging is a key tool in understanding the organization and function of cells, tissues and organisms, since it allows the visualization of dynamic processes within their native environment. However, in practice, the live-imaging of multi-layered tissues with different cell types often poses major challenges. Refractive index mismatches between tissue and surrounding medium result in spherical aberrations that misalign the optical paths and ultimately distort and attenuate the microscopic image. This effect increases with complexity and thickness of the specimen, making imaging in deep tissue layers difficult and technically demanding (*Richardson and Lichtman, 2015*).

Microscope optimization constitutes a first approach to optimize deep imaging. 2-photon microscopy greatly improves depth penetration by excitation with low scattering, near-infrared wavelengths (*Helmchen and Denk, 2005*). However, 2-photon microscopy cannot alleviate spherical aberration effects (*Richardson and Lichtman, 2015*). These can be partially compensated by the recent introduction of adaptive optics microscopes (*Booth, 2014*), yet at the cost of reduced image acquisition rates and the need for intense excitation light. A second approach to improving depth penetration is the direct adjustment of the refractive indexes (RI) of sample and environment (*Richardson and Lichtman, 2015*). Indeed, recently developed optical clearing techniques can render tissues effectively transparent by equilibrating refractive index heterogeneity within biological samples (*Chung et al., 2013*; *Hama et al., 2015*). Unfortunately, these protocols remain limited to fixed specimens due to their reliance on harsh mounting conditions and/or toxic chemicals (*Richardson and Lichtman, 2015*).

## Results

Towards the goal of developing an RI matching medium for live-imaging, we searched for compounds that combine high water solubility as prerequisite for dilution into regular culture media,

**eLife digest** Light microscopy is a key tool in biomedical research. For perfect images, light needs to be able to pass through the sample, the material (or "mounting medium") that holds the sample in place, and finally the image-detecting equipment in a straight line. However, in practice, light rays often deviate away from this line because they move at different speeds in different materials; how much the speed of light changes is related to a property called the refractive index of the material. This is exactly the effect that causes a stick stuck into water to look bent at the water's surface. In light microscopy, mismatches in refractive index significantly reduce quality of the images that can be obtained.

Live specimens are particularly challenging to image because different specimens have very different refractive indices compared to the mounting medium, which holds specimens in place but must also keep them alive. Although the addition of chemical compounds can theoretically match the refractive index of the mounting medium to that of the specimen, this approach has so far not been practical because such manipulations tend to kill the specimen. An important challenge has therefore been to identify a compound that can adjust, or "tune", the refractive index of mounting media over a wide range, yet without harming the specimens.

Now, Boothe et al. have identified a chemical called Iodixanol as an ideal and easy to use supplement for tuning the refractive index of water-based live imaging media. Adding Iodixanol to the mounting media did not appear to have any toxic effects on cell cultures, developing zebrafish embryos or regenerating planarian flatworms. Importantly, Boothe et al. found that Iodixanol significantly improved the quality of the images collected from all of these different specimens.

It is important to stress that Iodixanol does not change the refractive index of the sample or cancel out refractive index differences within the sample – so it cannot render opaque specimens transparent. Nevertheless, Iodixanol supplementation is a simple and affordable technique to improve image quality in any live imaging application without having to resort to more expensive and highly specialized microscopes.

dilution-dependent RI tuning for effectiveness with a wide range of specimens and finally low toxicity as crucial requirement for live-imaging compatibility. The compound Iodixanol, which was originally developed as an intravenous X-ray contrast agent (*Albrechtsson et al., 1992*) and widely used in density gradient applications (*Bettinger et al., 2002*), appeared to have many of the desired properties. Commercially available under the brand name OptiPrep<sup>(TM)</sup>, Iodixanol is optically clear and displays a high refractive index of 1.429 as a 60% stock solution, likely at least in parts due to its high density. This value is close to the refractive index of popular fixed tissue clearing solutions such as FocusClear (RI 1.47) or CLARITY (RI 1.45) (*Richardson and Lichtman, 2015*), and Iodixanol has in fact been used in such protocols (*Ku et al., 2016*).

As first test of its principal suitability, we evaluated the physicochemical properties of Iodixanol solutions. As Iodixanol is highly water-soluble, simple dilution into aqueous solutions can be used to linearly tune the refractive index of the solutions between RI 1.333 – RI 1.429 (*Figure 1a*). For water, PBS and culture media of aquatic model organisms, the medium only minimally affected the refractive index at a given Iodixanol concentration (*Figure 1a*). Further, we found the temperature dependent change in refractive index (*Beysens and Calmettes, 1977*) of Iodixanol solutions to be minimal within physiologically relevant temperature ranges (*Figure 1b*), allowing the use of the same medium at multiple temperatures. Organisms are often immobilized in agarose for live imaging. Agarose polymerization was not prevented at any Iodixanol concentration and agarose concentrations in ranges used for specimen immobilization did not significantly affect the refractive index of Iodixanol solutions (*Figure 1c*), thus making Iodixanol compatible with agarose embedding protocols. A further important requirement especially for fluorescence-based live imaging applications is low autofluorescence. A spectral emission scan of Iodixanol solutions at the commonly used excitation wavelengths of 405, 488, 560 and 640 nm failed to reveal significant autofluorescence in comparison with PBS or highly diluted fluorescent beads as negative or positive controls, respectively (*Figure 1d*, *Figure 1—figure supplement 1*). pH buffering capacity is a further important

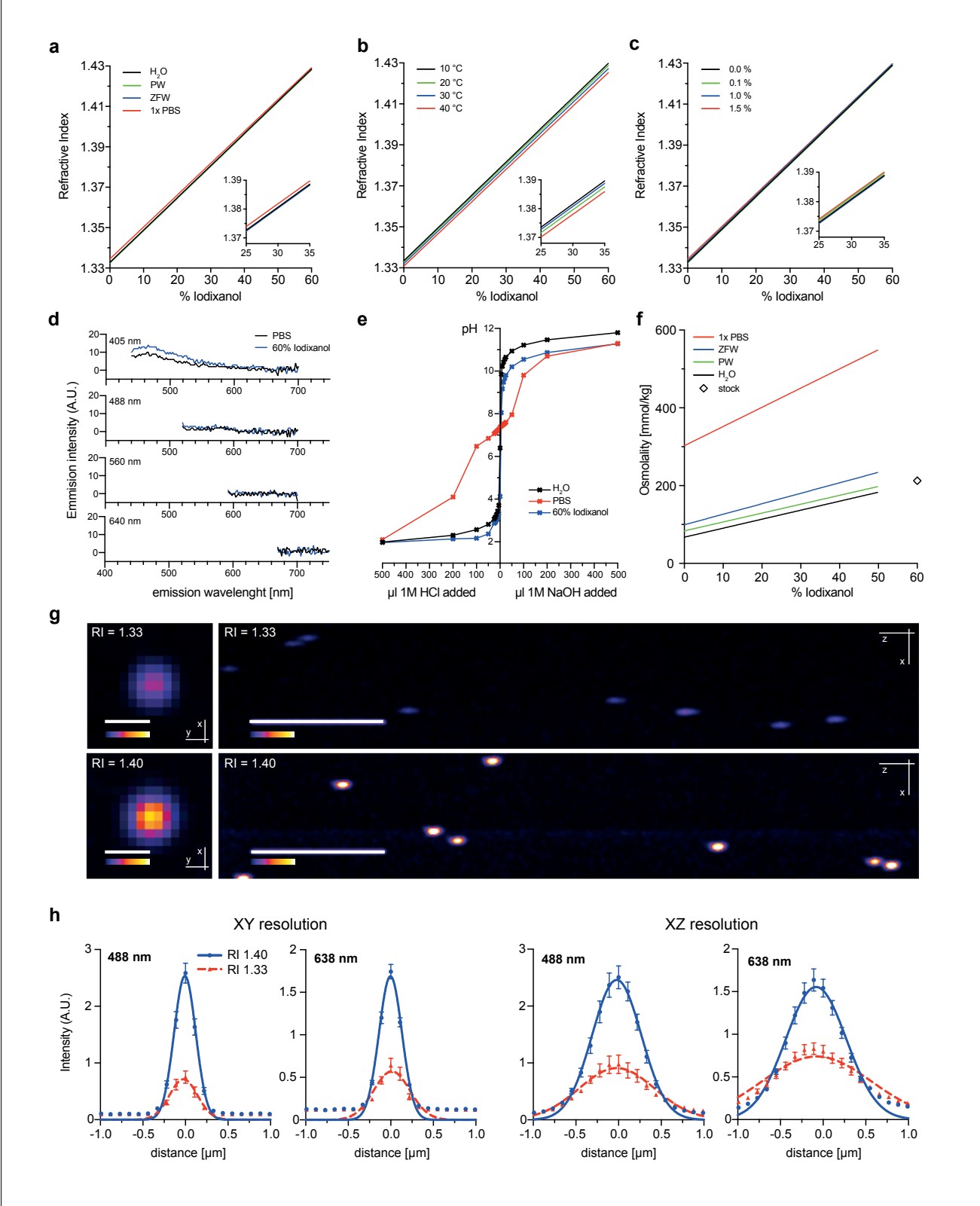

**Figure 1.** Physicochemical properties of the refractive index matching agent Iodixanol. (**a**). Solvent dependency of the refractive index of Iodixanol. (**b**) Temperature dependency of the refractive index of Iodixanol solutions. Water was used as a solvent. (**c**) The refractive index of Iodixanol gels at various agarose concentrations. (**a–c**) Inset diagrams show a magnified region of the respective data set. Measurements were taken at 10% Iodixanol concentration increments as technical triplicates. In all cases a linear regression curve fit was applied to the series and the coefficient of determination is

*Figure 1 continued on next page*

*Figure 1 continued*

in all cases $r^2 > 0.999$. Standard deviations ($\sigma < 0.01\%$ in all cases) and data points were omitted for simplicity. See *Figure 1—source data 1–3* for raw measurements. (d) Autofluorescene emission spectra of Iodixanol compared to PBS measured at indicated excitation wavelengths. Note that the detected signal is by orders of magnitudes lower than that of a positive fluorescent control, even at 405 nm excitation (*Figure 1—figure supplement 1*). (e) pH titration curve of 60% Iodixanol stock solution and indicated reference solutions. Starting volume = 50 ml. Data obtained from a single experiment. (f) Osmolality of Iodixanol solutions in various solvents. Measurements were taken at 10% Iodixanol concentration increments as technical triplicates and a linear regression curve fit was applied to the series. The coefficient of determination is in all cases $r^2 > 0.981$. Standard deviations ($\sigma < 0.5\%$ in all cases) and data points were omitted for simplicity. See *Figure 1—source data 4* for raw data. (g) 100 nm sub-diffraction sized beads imaged at 488 nm in unsupplemented aqueous 1% agarose (top panel) or in Iodixanol supplemented agarose tuned to the refractive index of the silicon immersion oil used for imaging (bottom panel). Lateral pictures (left) show a single optical plane while axial pictures (right) represent maximum projected y-stacks. Scale bars: lateral 0.5 µm, axial 10 µm. The colour scheme encodes relative intensity (brightest = white) and all image acquisitions were performed under identical microscope settings (h) Point spread functions of sub-diffraction sized beads as shown in (g). Quantified were peak intensity signal distributions from individual optical planes at indicated excitation wavelengths and direction (n = 20, error bars represent S.E.M). See *Supplementary file 1* for quantified resolutions. Abbreviations: PBS: phosphate buffered saline; PW: planarian water; RI: refractive index; ZFW: zebrafish water.

The following source data and figure supplement are available for figure 1:

**Source data 1.** Raw measurement values for solvent dependency of the refractive index of Iodixanol.

**Source data 2.** Raw measurement values for temperature dependency of the refractive index of Iodixanol solutions.

**Source data 3.** Raw measurement values for the refractive index of Iodixanol gels at various agarose concentrations.

**Source data 4.** Raw measurement values for the osmolality of Iodixanol solutions in various solvents.

**Figure supplement 1.** Autofluorescence measurements of 60% Iodixanol compared to a highly dilute fluorescent bead solution (0.04% solids) as positive controls at indicated excitation wavelengths.

consideration for potential media supplements. pH titration curves demonstrate that Iodixanol solutions have no significant pH buffering capabilities within the physiological relevant pH range of pH 4 – pH 9, especially in comparison with PBS as classical physiological buffer (*Figure 1e*). In fact, Iodixanol is only a slightly stronger acid than water (*Figure 1e*). Finally, many optical clearing agents, such as ScaleA2, have a high intrinsic osmolality that makes the reagent intrinsically live specimen incompatible (*Ke et al., 2013*). 60% OptiPrep stock solution displays an osmolality of $212 \pm 2$ mmol/kg, which is below the typical 290–300 mmol/kg of vertebrate cell culture media (*Figure 1f*). Further, we measured a linear increase of media osmolality across a dilution series with increasing Iodixanol concentrations (*Figure 1f*). This means that the contribution of Iodixanol to overall media osmolality can be offset by a corresponding decrease in media salt concentration (e.g., NaCl).

To assess the optical effects of Iodixanol supplementation on image quality, we quantified the point spread functions of sub-diffraction sized fluorescent beads using a high NA 1.35 silicon oil immersion objective. As expected, tuning of the refractive index of the bead solution to that of the used silicon immersion oil (RI = 1.40), greatly improved both the lateral and axial image resolution compared to controls mounted in conventional aqueous media (RI = 1.33; *Figure 1g,h*; *Supplementary file 1*). Overall, the physicochemical properties of Iodixanol are therefore ideally suitable for refractive index tuning of live imaging media.

However, toxicity is a further crucial concern in live imaging applications. We therefore quantitatively assessed the health of a range of typical specimens under extended Iodixanol exposure. We first measured the growth rates of human HeLa cell cultures exposed to various concentrations of Iodixanol 24 hr after seeding. Our quantitative measurements failed to detect any Iodixanol concentration dependent effects on HeLa cell proliferation or cell death up to three days after plating, even at the highest tested concentration of 30% Iodixanol (*Figure 2a*, *Figure 2—figure supplement 1*). Importantly, a concentration of 30% Iodixanol (RI = 1.380) is higher than the optimal Iodixanol concentration required for HeLa cells. In absence of any toxicity indications, we carried out all subsequent toxicity assessments at the optimal Iodixanol concentration for the respective specimens

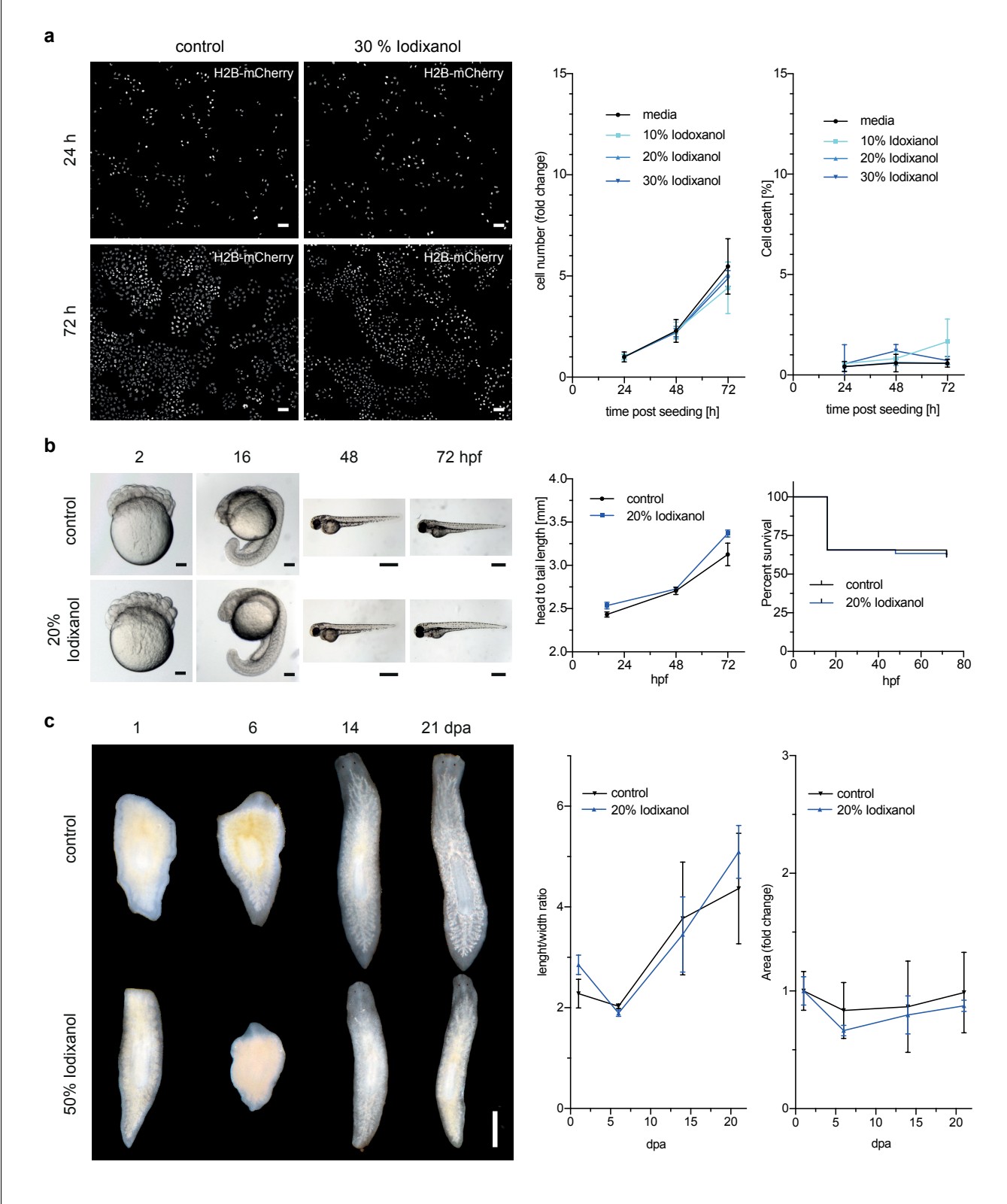

**Figure 2.** Iodixanol is live specimen compatible. (a) Iodixanol does not affect growth and cell death levels in cultured HeLa cells. Left: Representative low-resolution images of the constitutively expressed nuclear marker H2B-mCherry at indicated incubation times and media conditions. Scale bar = 50 μm; Right: Quantification of cell numbers (number of nuclei) and dead cells (DRAQ7 positive nuclei) at the indicated time points and Iodixanol concentrations. Iodixanol was applied 24 hr post seeding and measurements were normalized to the 24 hr time point in order to compensate

*Figure 2 continued on next page*

*Figure 2 continued*

fluctuations in plating density. n = 3; See *Figure 2—figure supplement 1* for a complete data representation. (b) Iodixanol does not affect developmental growth or survival of dechorionated zebrafish embryos. Left: Representative images of developing embryos at the indicated time points (hpf = hours post fertilization) and media conditions. N = 5; Scale bars = 100 µm at 2 and 16 hpf, 500 µm at 48 and 72 hpf. Right: Quantification of body length and survival rate at the indicated time points and media conditions. The initial drop in the survival curves is an effect of dechorionation. N = 30; (c) Iodixanol does not affect regeneration of the planarian head or body proportions. Left: Representative images of regenerating *Dendrocoelum lacteum* amputation fragments at the indicated time points (dpa = days post amputation) and under the indicated media conditions. Anterior is always up, Scale bar = 500 µm; Right: Quantification of length/width ratio and projected area at the indicated time points and media conditions. Measurements were normalized to the 0 time point in order to compensate initial size differences between tissue pieces. N = 3; (a–c) Error bars represent S.E.M. p>0.05 in all cases: (a) one way ANOVA (b, c) paired t-test.

The following figure supplement is available for figure 2:

**Figure supplement 1.** Representative low-resolution images of HeLa cell cultures exposed to the indicated Iodixanol concentrations at the indicated time points.

(please see Materials and methods and *Figure 4—figure supplement 2* for a guide on how to determine a specimen's optimal Iodixanol concentration).

We next assessed Iodixanol exposure effects on development by exposing de-chorionated zebrafish embryos to the optimal concentration of 20 % w/v Iodixanol. At 72 hr post fertilization, all embryos developing in Iodixanol displayed normal motility, muscle contractions and body pigmentation. Further, we found survival rates and the head to tail length as measure of developmental growth to be indistinguishable from controls, indicating that Iodixanol exposure over three days of development neither overtly affected development nor survival of zebrafish embryos (*Figure 2b*).

To assess potential long-term effects of Iodixanol exposure on dynamic tissue-level processes, we mounted regeneration-competent tissue fragments of planarian flatworms (*Rink, 2013*) in 50 % w/v Iodixanol. Remarkably, even after 3 weeks of continuous exposure to a high concentration of Iodixanol, the specimens were healthy, had regenerated morphologically normal heads and succeeded in restoring normal body plan proportions as quantified by length to width ratio and projected area in a manner indistinguishable from controls (*Figure 2c*). Collectively, these results establish that Iodixanol supplementation minimally impacts survival and growth of cell cultures, embryonic development or tissue turn-over and regeneration in intact animals, thus largely alleviating sample toxicity concerns.

We therefore assessed the though-after improvements in live image quality obtainable via Iodixanol refractive index tuning. As reference point we used a current state of the art spinning disc confocal microscope with silicone immersion oil objectives. The refractive index of silicone oil, RI = 1.406 closely matches typical live specimens and its introduction has afforded a substantial improvement in live imaging quality (*York et al., 2012*). We started our investigations at the smallest functional scale by imaging clusters of cultured primary zebrafish cells. In unsupplemented mounting media, the structure of nuclear chromatin was indiscernible in cells located 'behind' the first layer along the z-axis. Tuning the mounting media RI to 1.362 reduced the degradation of image resolution for such cells, demonstrating improvements in high resolution imaging of multi-layered cell culture applications (*Figure 3a*, *Figure 3—figure supplement 1*). Organoids, which are currently emerging as an important ex vivo model of organ development and function (*Simian and Bissell, 2017*), represent an imaging challenge at a larger functional scale. Human cerebral organoids appear opaque due to the optical density of neuronal tissues (*Figure 3b*) (*Lancaster and Knoblich, 2014*). Consequently, conventional single photon microscopy cannot penetrate significantly beyond 20 µm depth (*Figure 3b*). By mounting organoids (67 days aged) in Iodixanol supplemented culture media (RI = 1.363), we doubled the penetration depth to ~40 µm as a consequence of improved signal to noise ratios at depth (*Figure 3b*). Iodixanol supplementation thus improves depth penetration in organoid imaging.

Zebrafish embryos are a popular vertebrate development model system because of their optical transparency (*Vascotto et al., 1997*), yet the segmentation and tracking of cells beyond 100 µm in depth is still a challenge in embryos mounted in culture media (RI = 1.333, *Figure 4—figure supplement 1*). To quantitatively assess the effect of Iodixanol supplementation on resolution and thus

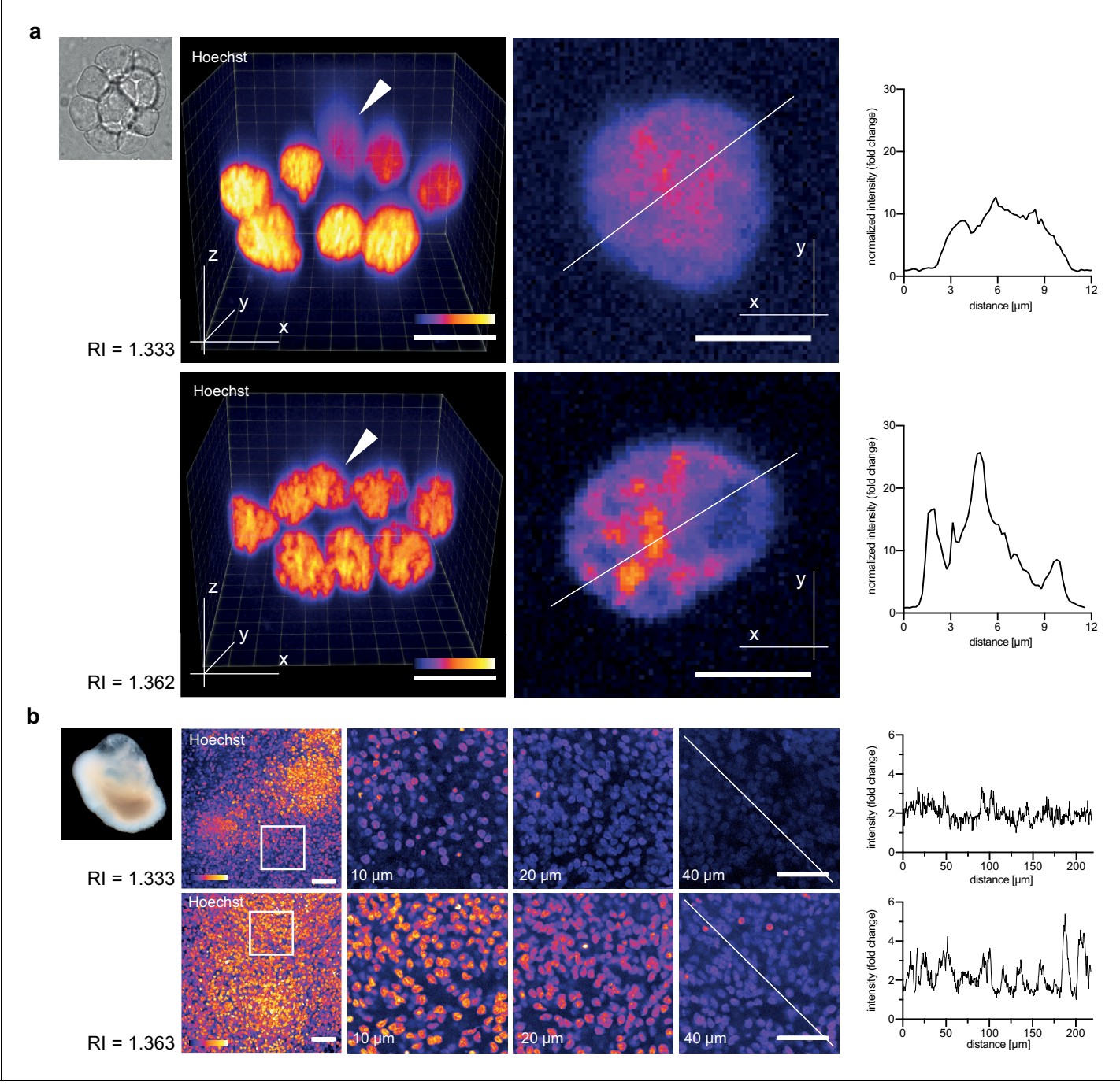

**Figure 3.** Refractive index tuning with Iodixanol improves live-imaging of tissue culture systems. (**a**). Effects of Iodixanol supplementation on live imaging of primary zebrafish cell cultures. Top Left: Brightfield image of a representative cluster of primary zebrafish embryonic cells, approximatly 50 μm in diameter. Centre panel: Images of cell clusters stained with the nuclear dye Hoechst 33342. Left column: 3D-reconstruction of representative multi-layered cell clusters, imaged in control media (RI = 1.333, top row) or in refractive index matched media (RI = 1.362, bottom row) under identical imaging conditions. The arrowheads indicate representative deep layer nuclei that are further shown as 2D optical XY-section in the right column. Graphs: Intensity profiles along the solid lines indicated in the respective xy-section image. The flatter and lower intensity profile in the control condition (top) quantitatively documents a loss of chromatin structure fine detail in deep nuclei, which is preserved by Iodixanol supplementation (bottom). Scale bars = 3D: 10 μm and 2D: 5 μm See *Figure 3—figure supplement 1* for orthogonal sections. (**b**) Effects of Iodixanol supplementation on live imaging of human cerebral organoids. Top left: Dark field image of a representative human cerebral organoid approximately 2 mm in diameter. Centre panel: Human cerebral organoids at culture day 67 stained with the nuclear dye Hoechst 33342. Centre panel: 3D-imaging of organoids, mounted either in standard media (RI = 1.333, top row) or in refractive index matched media (RI = 1.363, bottom row) under identical imaging conditions. Left column: Maximum projections of representative z-stacks. The white frame indicates the region shown to the right as optical xy-sections

*Figure 3 continued on next page*

*Figure 3 continued*

at the indicated tissue depth. The solid white line across the deepest section traces the course of the pixel intensity profile shown to the right. The flatter and lower intensity profile in the standard condition (top) quantitatively documents the loss of nuclear signal at 40 µm depth, while Iodixanol supplementation (bottom) still allows nuclei detection at that depth. Scale bars = 50 µm. The color scheme encodes relative intensity (brightest = white).

The following figure supplement is available for figure 3:

**Figure supplement 1.** Zebrafish primary cell culture.

penetration depth, we imaged embryos injected with sub-diffraction sized fluorescent beads. The quantification of lateral point spread functions between controls and embryos mounted in media tuned to a refractive index of RI 1.363 revealed an improvement of lateral resolution (792 ± 28 nm) compared to specimens mounted in regular media (918 ± 50 nm) at a distance of 150 µm from the coverslip (*Figure 4a*). We found that the resolution benefit of refractive index tuning increases with the distance of the object plane to the coverslip (*Figure 4a*, *Supplementary file 1*), as expected from the increasing impact of spherical aberrations with increasing distance to the objective. Overall, RI tuning of the embedding media to RI 1.363 allowed segmentation of nuclei up to 300 µm in depth, thus demonstrating a substantial improvement of deep tissue imaging in developing zebra-fish embryos (*Figure 4—figure supplement 1*).

As final imaging challenge, we chose planarian flatworms. Although these animals are widely studied as models of whole body regeneration, live imaging of planarian regeneration has so far not been possible. Even unpigmented species like *Dendrocoelum lacteum* (*Liu et al., 2013*) are optically highly opaque, such that live imaging is largely restricted to the outermost cell layer (the epithelium; *Figure 4b*). By tuning the refractive index of the embedding medium to RI 1.412, we could partially compensate the opaque appearance of the specimen and significantly improve both signal detection and the overall signal to noise ratio in deeper cell layers (*Figure 4b*). Together with the lack of overt effects on regeneration (*Figure 2c*), Iodixanol supplementation therefore brings within reach the live-imaging of cell dynamics during planarian regeneration.

## Discussion

Overall, our results establish Iodixanol supplementation as a simple, versatile and effective method for refractive index tuning in live imaging applications. We show that the reduction of spherical aber-rations between sample and mounting media by refractive index tuning provides substantial improvements in achievable imaging depth in planaria, zebrafish and human organoids, as well as improved spatial resolution in cell culture applications. Refractive index tuning with Iodixanol there-fore enables alignment of an important aspect of the optical axis in live specimens that could so far only be compensated in fixed specimens. What Iodixanol supplementation cannot correct for are refractive index differences within the specimen, such as between neighboring cells or between organelles and surrounding cytoplasm. Such effects are likely responsible for the fact that planarians and organoids appear optically opaque despite lacking pigmentation. Even though Iodixanol can therefore not achieve the *in-toto* RI matching of fixed tissue protocols (*Richardson and Lichtman, 2015*), our results nevertheless demonstrate substantial imaging improvements even in the case of opaque specimens. Overall, we expect that refractive index tuning by Iodixanol supplementation represents a broadly useful addition to the tool kit of live-imaging applications, all the way from cells to tissues and organisms.

## Material and methods

### Reagents

Iodixanol/OptiPrep was purchased as a 60% w/v stock solution from Sigma (Cat No. D1556). Planar-ian water contained 1.6 mM NaCl, 1 mM CaCl$_2$, 1 mM MgSO$_4$, 0.1 mM MgCl$_2$, 0.1 mM KCl, 1.2 mM NaHCO$_3$. Zebrafish medium contained 0.3x Danieu's (17.4 mM NaCl, 228 µM KCl, 122 µM

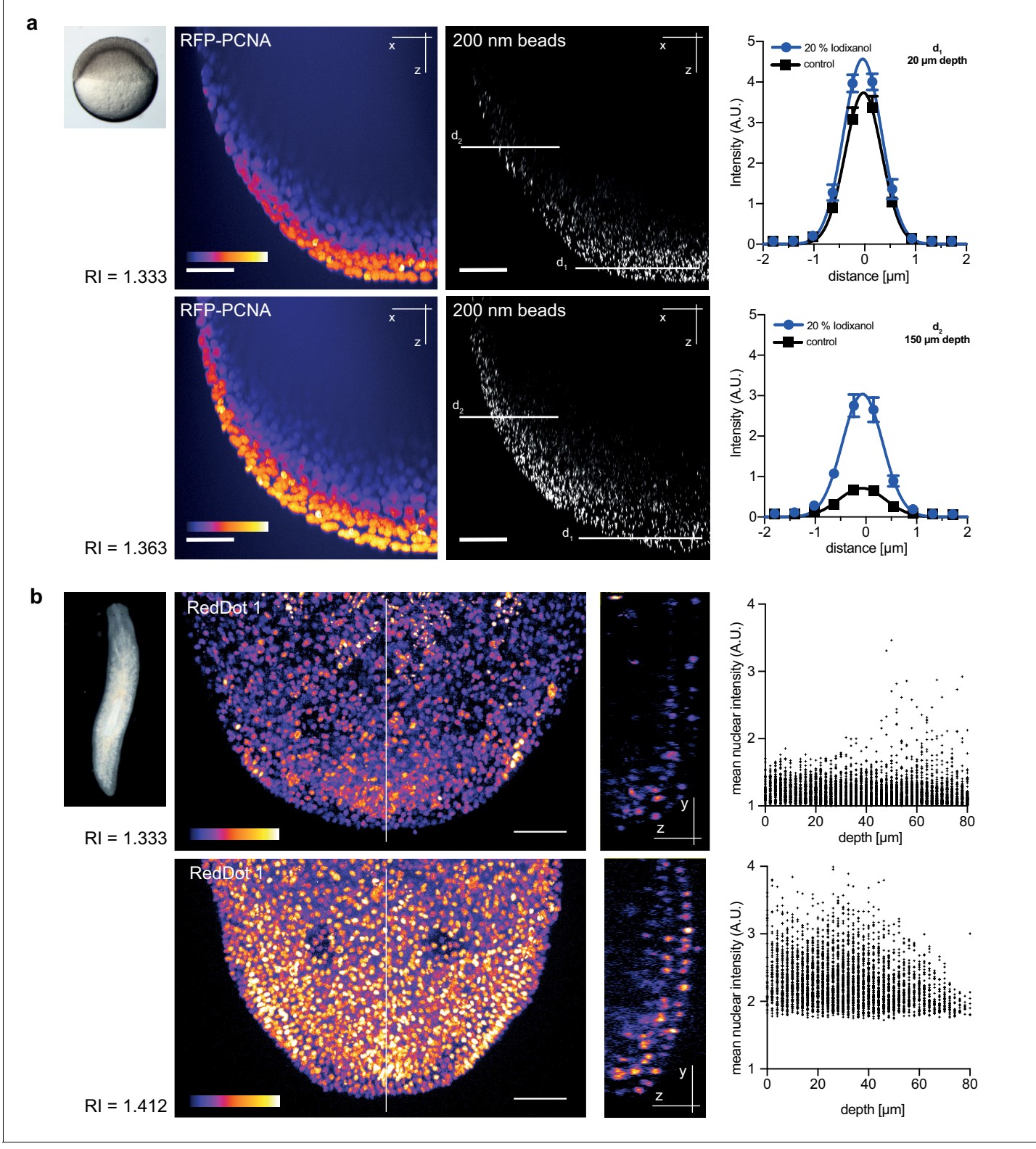

**Figure 4.** Refractive index tuning with Iodixanol improves model organism live-imaging. (**a**). Effects of Iodixanol supplementation on zebrafish embryo live imaging. Top left: Stereoscopic image of a dome stage zebrafish embryo of approximately 700 μm diameter. Centre panel: Zebrafish embryos expressing RFP-PCNA were injected at the single cell stage with 200 nm fluorescent sub-diffraction sized beads and imaged at dome stage (4 hpf). Images (left column: RFP-PCNA, right column: beads) represent 50 μm thick y-maximum projections of embryos imaged in regular media (RI = 1.333,
*Figure 4 continued on next page*

*Figure 4 continued*

top row) or in Iodixanol supplemented media (RI = 1.363, bottom row). Scale bars = 50 µm. The graphs to the right depict the quantification of point spread functions of individual beads (N = 20) at shallow (top) or deep (bottom) imaging depth, each for control and Iodixanol mounted specimens as per the indicated color scheme. The position of the analysed planes is indicated in the bead images to the left. The quantification of the width of the point spread function at half-maximal amplitude (See *Supplementary file 1* for numerical results) reveals a significant increase in resolution in deep sections. (**b**) Effects of Iodixanol supplementation on planarian live imaging. Top left: Dark-field image of a specimen of the planarian flatworm *Dendrocoelum lacteum* approximately 4 mm in length. Centre panel: Live *Dendrocoelum lacteum* were stained with the nuclear marker RedDot1 and mounted in control media (RI = 1.333, top) or media Iodixanol-tuned to RI = 1.412 (bottom). The large images represent z-maximum projections of image stacks in the head region, with the solid line indicating the position of the single-plane orthogonal yz-section shown to the right. The scatter plots of mean nuclear intensity versus depth to the right quantitatively document an improved signal return upon Iodixanol supplementation, especially from deeper tissue layers. Scale bars = 50 µm. The color scheme encodes relative intensity (brightest = white).

The following figure supplements are available for figure 4:

**Figure supplement 1.** Improved nuclear segmentation in deep tissue layers by Iodixanol supplementation.

**Figure supplement 2.** Schematic guidelines for determining the optimal Iodixanol concentration for a given specimen.

---

MgSO4*7$H_2$O, 262 µM Ca($NO_3$)$_2$, 1.5 mM HEPES). Dissociated zebrafish cells were cultured in 1x Dulbecco's PBS (DPBS) with 0.8 mM $CaCl_2$. Organoids were cultured in Differentiation Medium with vitamin A (125 ml DMEM/F12, 125 ml Neuralbasal, 1.25 ml N2 supplement, 2.5 ml B27 + vitamin A supplement, 62.5 µl insulin, 2.5 ml Glutamax supplement, 1.25 ml NEAA-MEM and 2.5 ml penicillin-streptomycin) according to Lancaster et al (*Lancaster and Knoblich, 2014*). Low gelling temperature SeaPlaque agarose (Lonza, Cat No. 50100) was used for sample embedding. For imaging, samples were mounted in 35 mm No. 1.5 glass bottom dishes (MatTek, Cat No P35G-1.5–14-C).

For the determination of optical resolution in vitro 0.1 µm TetraSpeck fluorescent beads (Thermo Fisher Scientific, Cat No.: T7279) mounted in 1% SeaPlaque agarose were used. The resolution was determined in vivo with 0.2 µm FluoSpheres (Thermo Fisher Scientific, Cat.: F8807).

## Determination of iodixanol's physical properties

Refractive indexes were measured at 20°C unless otherwise indicated. Measurements of the refractive index were performed on a Rudolph Research Automatic Refractometer J457 at a wavelength of 589.3 nm. Each measurement was performed as a technical triplicate and refractive indexes were measured at 0%, 10%, 20%, 30%, 50% and 60% final Iodixanol content.

Osmolality measurements were performed with a Wesco Vapro Osmometer as technical triplicates. For each Iodixanol dilution series the instrument was independently calibrated. The osmolality was measured at 0%, 10%, 20%, 30%, 50% and 60% final Iodixanol content.

pH titration was performed with a freshly calibrated digital PHM210 pH meter (Radiometer Analytical). The 1M HCl and 1M NaOH titration were carried out in separate experiments. In both experiments 50 ml of the indicated solution were titrated by subsequently adding 5, 10, 15, 20, 25, 50, 100, 200 and finally 500 µl of acid or base. Measurements were taken once the pH meter indicated a stable measurement.

Further information on Iodixanols physical properties (such as density and viscosity) can be found on the product information sheets of the respective commercial vendors (an extensive description is provided by Alere Technologies: https://goo.gl/I4owRU).

## Determining the optimal iodixanol concentration for live imaging

Which concentration of Iodixanol ($c_{\%Iodixanol}$) needs to be used is highly specimen dependent. If the refractive index of the sample is known the refractive index of the media ($RI_{media}$) should be adjusted accordingly simply by Iodixanol dilution:

$$c_{\%\text{Iodixanol}} \approx \frac{(RI_{media} - 1.333)}{0.0016}$$

(equation based on data from *Figure 1a*). When the refractive index of the sample is unknown an Iodixanol concentration titration should be performed. In this method introduced by Oster *et al.*,

samples are incubated in various concentrations of Iodixanol and observed with phase contrast microscopy (*Oster, 1956*). A loss of contrast between sample and media results from a match of refractive indexes and thus experimentally indicates the target Iodixanol concentration (*Figure 4— figure supplement 2*)

## Live sample preparation

HeLa 'Kyoto' cells stably expressing H2B-mCherry were described previously (*Neumann et al., 2010*) and obtained from the Ellenberg group at the European Molecular Biology Laboratory Heidelberg. HeLa 'Kyoto' cells are not included in the Register of Misidentified Cell Lines v 8.0 curated by the International Cell Line Authentication Committee (*Capes-Davis et al., 2010*). The cell line was authenticated using Multiplex Cell Authentication by Multiplexion (Heidelberg, Germany) as described (*Castro et al., 2013*). The SNP profiles matched known profiles or were unique. Mycoplasma tests with negative results for contamination were performed using the VenorGeM mycoplasma detection kit (Sigma-Aldrich, Cat No. MP0025). HeLa cells were cultured at 37C and 5% CO2 in High glucose GlutaMAX DMEM media (Thermo Fischer Scientific, Cat No.: 10566016) supplemented with 10% (v/v) heat inactivated FBS, 100 µg/ml Penicillin/Streptomycin and 0.5 µg/ml Puromycin as a selection agent. For monitoring cell proliferation and death 700 cells were seeded per well into a 384 well plate (Greiner Bio-One, Cat No.: 781096). 24 hr post seeding media was replaced with 0%,10%, 20% or 30% Iodixanol supplemented standard culture media additionally supplemented with 1.5 µM DRAQ7 (Cell Signaling Technologies, Cat No.: 7406S) as a cell death marker. Due to the high density of Iodixanol, plates were incubated upside down between image acquisitions. Imaging was carried out every 24 hr with the plate being in an upright position (see below).

*Dendrocoelum lacteum* were cultured in planarian water at 13°C and were fed weekly with calf liver paste. Prior imaging experiments animals were starved for 2 weeks. To stain planarian nuclei, animals were incubated for 12 hr with 2x RedDot1 (Biotium, Cat No.: 40060) and 1% (v/v) DMSO in planarian water. Prior mounting, animals were anesthetized and relaxed for 1 hr by supplementing planarian water with 0.0097% w/v Linalool (Sigma, Cat No. L2602). Animals' mucus was removed by a 5 min incubation in 0.5% w/v pH neutralized N-Acetyl-L-cysteine (Sigma, Cat No. A7250). Subsequently, animals were mounted in 1.5% SeaPlaque agarose dissolved in planarian water supplemented with 0.0097% Linalool. RI matched media had a final 50% Iodixanol content.

Zebrafish embryos were kept according to standard conditions. Embryos of wild type (TLAB) and transgenic (Tg(bactin:RFP-pcna)) fish, the latter a generous gift of Caren Norden, were dechorionated by pronase treatment and maintained at 28°C in 0.3X Danieu's medium diluted in distilled water and Iodixanol as indicated. Embryos were mounted in hanging drops of liquid mounting medium in ibidi glass bottom dishes (35 mm diameter, 0.17 mm coverslip), and inverted and submerged in liquid medium for imaging.

For zebrafish cell culture, embryos were dissociated into individual cells in 55 mM NaCl, 1.75 mM KCl, 1.25 mM NaHCO$_3$, 10% glycerol solution by vortexing in low retention micro-centrifuge tubes. The cell suspension was centrifuged (400 g, 1 min), supernatant aspirated and replaced with 110 mM NaCl, 3.5 mM KCl, 2.7 mM CaCl$_2$, 10 mM Tris/Cl (pH 8.5), 10% glycerol solution. After further centrifugation (400 g, 1 min), supernatant replaced with DPBS with 0.8 mM CaCl$_2$ added. This suspension was centrifuged (400 g, 1 min), the supernatant replaced with ~20 µl liquified agarose-based cell culture medium (liquified at 70°C and held at 38°C) and the cell pellet mechanically resuspended with a plastic micropipette tip. Liquid culture medium with suspended cells was transferred with the same micropipette tip onto the coverslip of an ibidi glass bottom dish. The still liquid mounting medium droplet was sandwiched with an additional 18 mm diameter round coverslip. After about 3 min the added coverslip was mechanically held down while applying 1 ml additional mounting medium. The imaging dish was then capped and sealed airtight with parafilm to prevent evaporation. Dissociation, mounting and imaging were carried out at room temperature without cooling or heating.

Control zebrafish cell culture medium was DPBS with 0.8 mM CaCl$_2$. RI-matched medium was prepared in several steps, starting with 0.7X Dulbecco-PBS, 20% Iodixanol, 0.8 mM CaCl$_2$. Osmolality was then lowered to 5 mOsm/kg of control medium as a reference by addition of distilled water and repeated osmolality measurement. The RI was then lowered to within 0.003 of the cytoplasmic RI (1.3615, determined by phase contrast microscopy [*Oster, 1956*]) by addition of control medium

and repeated RI measurement (refractometer, 25°C). Control and RI-matched media were divided into 2 ml aliquots in microcentrifuge tubes, supplemented with 0.7% UltraPure (Thermo Fisher Scientific, Cat No.: 16520050) low melting point agarose. Tubes were closed airtight and heated to 70°C for at least 1 hr, and could then be stored at 4°C for a month at minimum. For DNA staining, Hoechst 33342 stock (5 mg/ml) was spiked into mounting medium aliquots at 1:2000 (v/v) ratio before mounting.

Human cerebral organoids were generated from human iPSC line SC102A-1 (System Biosciences) and cultured according to previously published protocols with minor modification (*Lancaster and Knoblich, 2014* and *Camp et al., 2015*). The culture media was replaced with an 18% Iodixanol/media v/v solution 24 hr prior imaging to match the refractive index of the tissue. 2 hr prior imaging this solution was supplemented with 5 µg/ml Hoechst 34580 (Thermo Fisher Scientific, Cat No. H21486) to stain nuclei. Organoids were mounted in 1% SeaPlaque agarose for imaging.

The experiments performed with live samples did not require ethical approval according to German law.

## Imaging

Autofluorescence of Iodixanol and control solutions was measured on a Tecan Spark 20M plate reader. Fluorescence was measured at 405, 488, 560 and 640 nm excitation. The emission signals were detected by emission spectra scans starting at 440, 520, 592 and 670 nm respectively. Scans were performed in 2 nm intervals.

HeLa cell proliferation and death was monitored using a Cell Voyager 7000 spinning disc high throughput confocal system (Yokogawa Electric Cooperation). H2B-mCherry was excited with a 561 nm solid state laser and the emission signal was detected with a 600/37 nm bandpass filter. DRAQ7 was illuminated with a 640 nm solid state laser and emission was detected with a 676/29 nm bandpass filter. Imaging was performed with a 10x UPlSApo NA 0.4 air objective.

Fluorescent images of all other experiments were acquired on an Andor Revolution WD Borealis confocal spinning disc system. The Olympus IX83 stand was equipped with an Andor iXon Ultra 888 EMCCD for image acquisition. In vitro point spread functions were determined with an Olympus 100x NA 1.35 Sil UPlanSApo objective. For planarian, zebrafish embryo and organoid imaging an Olympus 30x UPlan SApo NA 1.05 Sil objective was used. Imaging of cultured zebrafish cells was performed with an Olympus 60x UPlan SApo NA 1.30 Sil objective. For Hoechst imaging a 405 nm laser diode was used in combination with 452/45 bandpass filter to detect the emission light. Green fluorescence of TetraSpeck beads was excited with a 488 nm laser diode and emission was collected with a 525/50 bandpass filter. RFP was illuminated with a 561 nm laser diode and the emission was detected with a 607/36 bandpass filter. RedDot1 was excited with a 640 nm laser diode and the emission was detected with a 685/40 bandpass filter. All filters were produced by Semrock.

In all comparisons between refractive index matched media to control conditions identical illumination (laser power) and detection parameters (exposure time) were used on identical hardware setups (objective, immersion silicone oil, filters, camera).

Regenerating *Dendrocoelum lacteum* were imaged on a Nikon AZ 100M widefield microscope stand equipped with dark field illumination and a Nikon AZ Plan Fluor 2x NA 0.2 lens mounted. Images were acquired with a Nikon Digital Sight DS-Fi1 camera.

Zebrafish embryo development was documented using a Leica M165C stereoscope equipped with a Leica 1x Plan apochromat NA 0.35.

Phase contrast imaging of zebrafish primary cell clusters was performed on a Zeiss Axioert 200M widefield microscope equipped with a Zeiss 20x Plan-Apochromat NA 0.75 objective. Images were recorded with a Diagnostics Instruments Spot RT camera.

Phase contrast imaging of HeLa cells was performed on a widefield Zeiss Observer Z1 microscope stand equipped with a Zeiss Axiocam MRm and a 40x LS Plan – NeoFluoar NA 0.6 lens.

## Image processing

Images were processed and analyzed with Fiji (*Schindelin et al., 2012*). 3D views were rendered with ClearVolume (*Royer et al., 2015*). Dynamic ranges, signal detection thresholds and object detection parameters were identically set when comparisons between refractive index matched and

control conditions were made. For better visualization of intensity levels the 'Fire' lookup table (LUT) was applied.

For segmenting nuclei in zebrafish embryos Fiji's implemented 'Otsu' adaptive thresholding method was used on the raw image stacks and particles larger than 100 pixels were considered nuclei. Mean intensities of the thresholded objects was measured and reported for each slice of the Z-stack.

For segmenting nuclei in planaria Fiji's implemented 'Moments' adaptive thresholding method was used on the raw image stacks and particles between 100 and 2000 pixels were considered nuclei. Mean intensities of the thresholded objects was measured and reported for each slice of the Z-stack.

To count live (H2B-mCherry) and dead (DRAQ7) nuclei of HeLa H2B-mCherry cells, images were automatically thresholded with Fiji's implemented 'Otsu' adaptive thresholding method for the respective channel. Thresholded objects larger than 100 pixels were counted as nuclei.

Intensities of beads for PSF determination or object intensities for the demonstration of signal to noise ratios in vivo were determined with Fiji's implemented 'Plot Profile' function along a previously defined line.

## Data processing

All experimental numerical data were processed and visualized with Graph Pad Prism software. For PSF determination, a Gaussian distribution function was fit to the raw measurements. The optical resolution was defined as the full-width at half maximum intensity of that function. Display figures were created using Adobe Illustrator software.

## Acknowledgements

TB and LH were supported by an ELBE postdoctoral fellowship from the Center for Systems Biology, Dresden. TB was further supported by an Add-on Fellowship in Systems Biology awarded by the Joachim Herz Stiftung. The authors want to thank the light microscopy facility (LMF) and the technology development studio (TDS) of the Max Planck Institute of Molecular Cell Biology and Genetics (MPI-CBG) for the outstanding technical support and advice throughout the progress of this project. The authors further want to thank the Tang Lab (MPI-CBG) for sharing their Tecan Spark 20 M plate reader with us.

## Additional information

### Funding

| Funder | Grant reference number | Author |
| --- | --- | --- |
| Max-Planck-Gesellschaft | Individual research support programs | Tobias Boothe<br>Lennart Hilbert<br>Michael Heide<br>Lea Berninger<br>Wieland B Huttner<br>Vasily Zaburdaev<br>Nadine L Vastenhouw<br>Eugene W Myers<br>David N Drechsel<br>Jochen C Rink |
| Center for Systems Biology, Dresden | ELBE Postdoctoral Fellowship | Tobias Boothe<br>Lennart Hilbert |
| Joachim Herz Stiftung | Add-on Fellowship in Systems Biology | Tobias Boothe |

The funders had no role in study design, data collection and interpretation, or the decision to submit the work for publication.

## Author contributions
TB, Conceptualization, Data curation, Formal analysis, Supervision, Funding acquisition, Investigation, Methodology, Writing—original draft, Project administration, Writing—review and editing; LH, Conceptualization, Data curation, Formal analysis, Funding acquisition, Investigation, Methodology, Writing—original draft, Writing—review and editing; MH, Conceptualization, Formal analysis, Investigation, Methodology, Writing—review and editing; LB, VZ, Methodology, Writing—review and editing; WBH, Writing—review and editing; NLV, EWM, Supervision, Writing—review and editing; DND, Conceptualization, Supervision, Writing—review and editing; JCR, Conceptualization, Formal analysis, Supervision, Investigation, Project administration, Writing—review and editing

## Author ORCIDs
Tobias Boothe, http://orcid.org/0000-0003-1925-0213
Wieland B Huttner, http://orcid.org/0000-0003-4143-7201
Nadine L Vastenhouw, http://orcid.org/0000-0001-8782-9775
Jochen C Rink, http://orcid.org/0000-0001-6381-6742

# Additional files

## Supplementary files
• Supplementary file 1. Quantitative Comparison of optical resolutions of the indicated optical systems and samples at various media refractive indexes. Errors are S.E.M, N = 20.

## Major datasets

The following dataset was generated:

| Author(s) | Year | Dataset title | Dataset URL | Database, license, and accessibility information |
|---|---|---|---|---|
| Boothe T, Hilbert L, Heide M, Berninger L, Zaburdaev V, Vastenhouw N, Huttner W, Myers G, Drechsel D, Rink J | 2017 | Data from: A tunable refractive index matching medium for live imaging cells, tissues and model organisms | http://datadryad.org/review?doi=doi:10.5061/dryad.83gp7 | Available at Dryad Digital Repository under a CC0 Public Domain Dedication |

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
