## [Decision Letter]

Thank you for submitting your article "A tunable refractive index matching medium for live imaging cells, tissues and model organisms" for consideration by *eLife*. Your article has been reviewed by two peer reviewers, and the evaluation has been overseen by a Reviewing Editor and Didier Stainier as the Senior Editor. The following individual involved in review of your submission has agreed to reveal his identity: Hari Shroff (Reviewer #1).

The reviewers have discussed the reviews with one another and the Reviewing Editor has drafted this decision to help you prepare a revised submission.

Summary:

There is a shared sentiment that the work is clever and the advance in terms of improving live 3D imaging is potentially very important, but the results are quite preliminary in terms of thorough quantitative validation, and thus it will require quite a bit of work to bring it up to the level of *eLife*.

We hope that the authors will take this on, because it would be very good for the community if the method got published in a general biology journal, rather than a microscopy or optics specialty journal, so that it would have a broader impact on the people who potentially will benefit most from it.

Essential revisions:

The key problems with the manuscript are that the method for obtaining the proper concentration of iodixanol is not delineated and there is not thorough quantitative documentation of either the effects of iodixanol on cell health and animal development or the improvements in resolution as a function of depth within a living specimen.

The ability of this method to be widely adopted critically depends on the ability of potential adopters to use this paper as a reference such that if they are using a specimen in tissue culture media or water and know the depth at which they want to image, they should have a chart or mathematical function to guide their choice of iodixanol concentration for imaging. This requires determination of the effects of iodixanol concentration on resolution as a function of depth for media and water. In addition, quantifying the effects of various iodixanol concentrations as a function of time of exposure (both time and light exposure) on cell division and death rates, as well as embryo growth and normal development is absolutely critical for validation of the method. Proper statistical analysis of these results will be of utmost importance.

The reviewers have excellent suggestions on how to carry these suggestions out, as well as several comments on other more minor issues that will benefit the strength of the conclusions (see below).

Reviewer #1:

In this manuscript, Rink and colleagues describe a simple method for minimizing spherical aberration that results due to refractive index mismatch at a boundary between the imaging media and live samples. The method uses a commercially available reagent, iodixanol, to index-match the media to the sample, thereby improving signal, resolution, and depth penetration in zebrafish cell culture, embryos, and planaria. Although the improvement is not dramatic, I still feel it is significant and, given the wide applicability of the method, the work seems appropriate for *eLife*. However, I have several concerns that need to be addressed before I can recommend publication.

First, it is unclear in general to me how the matching was determined. e.g. in Figure 2, how was the concentration of iodixanol determined? The authors are vague on this point, in general: 'In this and all subsequent applications, Iodixanol concentrations were carefully titrated until optimal imaging results were achieved and thus the media's refractive index was best matched to the refractive index of the specimen.' What criteria for 'optimal imaging' was used? The methods section fails to discuss this adequately, instead referring to [biophysical methods book reference]. I'm assuming the authors meant to cite a paper here – but even better would be careful, step-by-step directions as to how to determine the optimal RI so that others can replicate the work. I suspect such a protocol would go a long way to disseminating the method.

Second, the manuscript in general suffers from a lack of quantitative detail, instead qualitatively assuring us that samples were morphologically identical, divided at the same rate, etc as controls. Please provide quantitative evidence for these claims so that we can better assess them! Figure 2 shows pictures, but to my eye the planaria grown in iodixanol seem obviously smaller than the controls. We are told that the zebrafish embryos developed 'normally', but how – precisely – was 'normal' determined? The lack of quantitative detail extends to the description of the imaging improvement: 'largely abolished the degradation of image resolution for such cells'. More precision in claims like this would be helpful.

Another example – Figure 2 attempts to conclude that cultured cells are unaffected by 20% iodixanol. This conclusion is based on a visual inspection of cluster morphology under transmitted light. This is a wholly inadequate assessment of culture health. Indeed, throughout the first half of the manuscript, the authors could have and should have measured cell health using e.g. cell division rates, cell viability/vitality dyes, the levels of stress related proteins, etc. Establishing a non-cytotoxic but optically effective concentration of iodixanol (or at least an algorithm by which one can be found) is central to this work.

Furthermore all the viability work has been done on non-mammalian systems, while some of the imaging was done on human organoids. A viability study on mammalian cells would thus seem helpful.

Third, most of the imaging and biological viability work was done using 20% Iodixanol to generate a modest increase in refractive index (n=1.36). Presumably this is because the authors were trying to match the refractive index of bulk phase cytoplasm (n=1.3615, subsection “Live sample preparation”). However the refractive index of most tissues is considerably higher than that of pure cytoplasm – being influenced by, for example, the lipid and water fraction of the tissue.

This again raises the question of whether higher concentrations of iodixanol would further improve image quality and what concentrations can be tolerated by cells? What would greatly improve the quality of this manuscript is a titration of iodixanol concentration against optical aberration and biological stress/viability. Finding an optimal concentration that balances these two competing effects, or even just demonstrating an algorithm by which an optimal concentration can be found for a given biological system, would add tremendously to this work.

Reviewer #2:

Review of "A tunable refractive index matching medium for live imaging cells, tissues and model organisms" by Boothe et al.,.

This article presents a novel method for improving the resolution and useful penetration depth of fluorescence microscopy of live organisms. The authors define the rationale of the method as a well-known problem in the field of microscopy; namely, the mismatches of refractive indices (RIs) between the optical media through which the emitted fluorescence travels (cytosol, aqueous mounting media, glass coverslips, and objective immersion media) cause spherical aberration in the native images formed. This increased spherical aberration causes decreases in resolution, especially at depths far from the coverslip surface. The method presented in this paper serves as a solution to lower the effective spherical aberration of silicone oil immersion objectives by changing the RI of the media in which the live sample is mounted in order to better match the RI of the objective immersion medium. The authors demonstrate the usability of this RI-matching compound, Iodixanol, with a variety of live model organisms. The authors also demonstrate the increase in fluorescence intensity from live fluorescent organisms that are mounted in a RI-matched media, as compared with the normal aqueous media.

This article proposes a fantastic method for increasing the quality of fluorescent images of live organisms. Microscopists have been able to match RIs of fixed samples for years to increase image quality, but this is the first method detailing the possibility of RI matching in live samples with no observable effects on sample vitality. The authors comprehensively demonstrate the non-toxicity of Iodixanol on three common model organisms with varying degrees of complexity. The authors also effectively show that dilution of Iodixanol into traditional mounting media can easily tune the RI of the solution in a linear fashion, which will be useful to microscopists to easily implement with their own samples. The authors are very quantitative in their analysis of the increase in fluorescence intensity in a biological sample as a result of their RI matching method (Figure 3).

However, the authors fail to quantitatively support their assertion that their RI-matching method increases the practically achievable spatial resolution (in x/y or in z) of the system. There is no measurement or mention of the point-spread function of the objective, which is the true physical definition of the resolution of a microscope. Despite quantitatively showing that the intensity of fluorescence is higher at greater depths, the authors have not shown that the full-width at half maximum intensity (FWHM) of the objective's PSF remains more consistent with their RI-matching method than with traditional mounting methods. Without these data, the claims of providing "substantial improvements in achievable imaging depth" are not yet validated, since a deep image with a high signal-to-noise ratio (SNR) but low spatial resolution only ameliorates one of the two main issues that results from RI-mismatched spherical aberration. The data presented are encouraging and exciting as a live-cell microscopist, but ultimately incomplete at this point with respect to the conclusions.

Before recommending this paper for publication, I would make the following suggestions for experiments:

1) A quantitative comparison of spatial resolution in three dimensions between traditional media and Iodixanol-supplemented media, using an accepted resolution standard. I would suggest sub-diffraction limit multicolor fluorescent beads (100 nm diameter for high NA) embedded in 1-2% agarose. Fluorescent images of the beads should be acquired as three-dimensional stacks (similar parameters to the images presented in Figure 3) in order to quantify the effect that the Iodixanol has on the lateral (x/y) and axial (z) resolution of the objective. Since the beads will be below the diffraction limit, a three-dimensional gaussian intensity profile (PSF) can be fit to beads that are suspended at various depths away from the coverslip. In order to maximize the relevance of this quantification to most live-cell microscopists, I would recommend quantifying the PSFs using green (~500 nm) and red (~600 nm) fluorescence emission, as these wavelengths correspond to the two most common classes of biological fluorophores used in live-cell fluorescence microscopy (GFP and RFP). Additionally, I would recommend varying the concentration of Iodixanol in these experiments, as the refractive index matching will most likely correct for spherical aberrations at a subset of focal depths, rather than at every depth from the surface of the coverslip to the maximum working distance of the objective. I would recommend reporting the depth of the focal plane about which the spatial resolution is best for several useful concentrations of Iodixanol.

2) Perform a test of common drug solubility/activity in Iodixanol. While I am convinced of the non-toxicity of Iodixanol on vitality and growth of living organisms, many microscopists introduce inhibitory drugs into the mounting media to observe the effects that the drugs have on the organism at a sub-cellular level. Many of these microscopists would most likely not use Iodixanol in mounting media if Iodixanol inhibited the solubility or activity of their experimental drugs. I would suggest a commonly used DMSO-soluble drug, such as nocodazole, which has well-characterized organismal and sub-cellular phenotypes in many model organisms, and observe the differences (if any) of the drug's effect on the organisms' phenotype compared to traditional mounting media supplemented with the drug.

3) Measure and report the following physical parameters of Iodixanol at various concentrations: viscosity, autofluorescence, dispersive properties, the pH buffering capacity, and the refractive index at various wavelengths.

4) Use a higher-resolution, oil-immersion lens (numerical aperture {greater than or equal to} 1.4) with high refractive index immersion oil (n ~= 1.5). The authors mainly focus on silicone-immersion objectives, which offer greater working distance but lower native resolution and brightness than high-NA objectives. Microscopists that use these objectives also experience spherical aberration; in fact, the spherical aberration more sharply affects the resolution as a function of depth. I am interested to know if this RI-matching method improves the resolution and brightness of images using high-NA objectives at depths under 10μm from the coverslip.

---

## [Author Response]

*Summary:*

*There is a shared sentiment that the work is clever and the advance in terms of improving live 3D imaging is potentially very important, but the results are quite preliminary in terms of thorough quantitative validation, and thus it will require quite a bit of work to bring it up to the level of eLife.*

We would like to thank the reviewers and the editorial team for a constructive and competent *eLife* review round. Further, we greatly appreciate the positive assessment of our work and its importance for the live imaging community. We have addressed all shortcomings of the previous manuscript version as detailed below and are confident that the current version now indeed represents the intended quantitative evaluation of what Iodixanol supplementation can or cannot do. In short: A useful resource paper for the imaging community at large.

*We hope that the authors will take this on, because it would be very good for the community if the method got published in a general biology journal, rather than a microscopy or optics specialty journal, so that it would have a broader impact on the people who potentially will benefit most from it.*

*Essential revisions:*

*The key problems with the manuscript are that the method for obtaining the proper concentration of iodixanol is not delineated and there is not thorough quantitative documentation of either the effects of iodixanol on cell health and animal development or the improvements in resolution as a function of depth within a living specimen.*

*The ability of this method to be widely adopted critically depends on the ability of potential adopters to use this paper as a reference such that if they are using a specimen in tissue culture media or water and know the depth at which they want to image, they should have a chart or mathematical function to guide their choice of iodixanol concentration for imaging. This requires determination of the effects of iodixanol concentration on resolution as a function of depth for media and water. In addition, quantifying the effects of various iodixanol concentrations as a function of time of exposure (both time and light exposure) on cell division and death rates, as well as embryo growth and normal development is absolutely critical for validation of the method. Proper statistical analysis of these results will be of utmost importance.*

We agree and have added substantial new and quantitative data on all of the three key points:

1) Guide on determining optimal Iodixanol concentration: We have added a section to the supplemental material that provides a simple formula for specimens with no refractive index and a simple method for empirically determining the latter in cases where it is not known. (Figure 4—figure supplement 2).

2) Quantitative assessment of toxic side-effects on culture health and development: We have added substantial new and quantitative data on three different model systems. Specifically, we i) quantified growth dynamics and cell death in mammalian cell culture; ii) quantified head-tail length in developing zebrafish embryos as integrative measure of growth and development; iii) quantified body plan proportions and plan area as integrative measure of flatworm regeneration efficiency. None of these assays revealed significant discrepancies between controls and Iodixanol-treated specimens, which is why we can confidently state that Iodixanol supplementation has no overt toxicity in a wide range of model systems (Figure 2).

3) Quantitative demonstration of resolution improvements: We have added substantial new data, specifically in vitro and in vivo-determined point spread functions. The new data quantitatively demonstrates the improvement of image resolution by Iodixanol supplementation both in vitro (Figure 1) and in vivo (Figure 4).

*The reviewers have excellent suggestions on how to carry these suggestions out, as well as several comments on other more minor issues that will benefit the strength of the conclusions (see below).*

*Reviewer #1:*

*In this manuscript, Rink and colleagues describe a simple method for minimizing spherical aberration that results due to refractive index mismatch at a boundary between the imaging media and live samples. The method uses a commercially available reagent, iodixanol, to index-match the media to the sample, thereby improving signal, resolution, and depth penetration in zebrafish cell culture, embryos, and planaria. Although the improvement is not dramatic, I still feel it is significant and, given the wide applicability of the method, the work seems appropriate for eLife. However, I have several concerns that need to be addressed before I can recommend publication.*

*First, it is unclear in general to me how the matching was determined. e.g. in Figure 2, how was the concentration of iodixanol determined? The authors are vague on this point, in general: 'In this and all subsequent applications, Iodixanol concentrations were carefully titrated until optimal imaging results were achieved and thus the media's refractive index was best matched to the refractive index of the specimen.' What criteria for 'optimal imaging' was used? The methods section fails to discuss this adequately, instead referring to [biophysical methods book reference]. I'm assuming the authors meant to cite a paper here – but even better would be careful, step-by-step directions as to how to determine the optimal RI so that others can replicate the work. I suspect such a protocol would go a long way to disseminating the method.*

We agree with the reviewer that a description of how to determine the appropriate concentration of Iodixanol was lacking in the initial version of the manuscript and apologize for the missing reference. We now provide a formula which describes the desired concentration of Iodixanol when the RI of the specimen is known. For samples with unknown RI we describe a method for RI determination initially described by Oster et al., (Physical techniques in biological research: Cells and Tissues. Volume 3, 1956) which is based on phase contrast imaging. We generated a methodological scheme for these procedures in Figure 4—figure supplement 2 and described it in detail in the methods section and in the figure legend (Figure 4—figure supplement 2).

*Second, the manuscript in general suffers from a lack of quantitative detail, instead qualitatively assuring us that samples were morphologically identical, divided at the same rate, etc as controls. Please provide quantitative evidence for these claims so that we can better assess them! Figure 2 shows pictures, but to my eye the planaria grown in iodixanol seem obviously smaller than the controls. We are told that the zebrafish embryos developed 'normally', but how – precisely – was 'normal' determined?*

We now quantify survival rates and head/tail length of developing zebrafish as measure of developmental growth and find no difference between control and Iodixanol-treated specimens. These data along with representative images of both Iodixanol incubated and control fish are now shown in Figure 2. For planaria, we now quantified both the length to width ratio as well as the area of regenerating posterior body pieces in Iodixanol and control conditions. As now shown in Figure 2, there is no significant difference in the reestablishment of body plan proportions during regeneration in Iodixanol compared to control fragments. Please note that these animals have soft and highly deformable bodies and that cuts are made by hand, hence the comparatively large error bars and normalization of the data to the starting point.

*The lack of quantitative detail extends to the description of the imaging improvement: 'largely abolished the degradation of image resolution for such cells'. More precision in claims like this would be helpful.*

To further support our claim that Iodixanol supplementation increases image resolution we are now demonstrating quantitatively in vivo that point spread functions and therefore resolution are indeed improved especially in tissue layers further away from the coverslip. These data are now presented in Figure 4.

*Another example – Figure 2 attempts to conclude that cultured cells are unaffected by 20% iodixanol. This conclusion is based on a visual inspection of cluster morphology under transmitted light. This is a wholly inadequate assessment of culture health. Indeed, throughout the first half of the manuscript, the authors could have and should have measured cell health using e.g. cell division rates, cell viability/vitality dyes, the levels of stress related proteins, etc. Establishing a non-cytotoxic but optically effective concentration of iodixanol (or at least an algorithm by which one can be found) is central to this work.*

*Furthermore all the viability work has been done on non-mammalian systems, while some of the imaging was done on human organoids. A viability study on mammalian cells would thus seem helpful.*

We have added the requested quantitative data on a mammalian cell culture system and have quantified growth and death rates of cultured human HeLa cells under a range of Iodixanol concentrations. As now shown in Figure 2 and Figure 2—figure supplement 1, we could not observe any concentration dependent effects of Iodixanol on cell proliferation or cell death, which now quantitatively demonstrates a lack of adverse effects on cell growth.

*Third, most of the imaging and biological viability work was done using 20% Iodixanol to generate a modest increase in refractive index (n=1.36). Presumably this is because the authors were trying to match the refractive index of bulk phase cytoplasm (n=1.3615, subsection “Live sample preparation”). However the refractive index of most tissues is considerably higher than that of pure cytoplasm – being influenced by, for example, the lipid and water fraction of the tissue.*

*This again raises the question of whether higher concentrations of iodixanol would further improve image quality and what concentrations can be tolerated by cells? What would greatly improve the quality of this manuscript is a titration of iodixanol concentration against optical aberration and biological stress/viability. Finding an optimal concentration that balances these two competing effects, or even just demonstrating an algorithm by which an optimal concentration can be found for a given biological system, would add tremendously to this work.*

First, this point was the second reason for measuring the growth of HeLa cells under a range of Iodixanol concentrations. As we could not detect any concentration dependent effect of Iodixanol on cell proliferation and death (Figure 2 and Figure 2—figure supplement 1), these data indicate that there are no strong penalties associated with increasing Iodixanol concentrations, which is not so surprising given that this compound has been used for vital imaging of the human brain. Importantly, we used Iodixanol concentrations of up to 30%, which is above the optimal HeLa target concentration of ≈ 20% (see Figure 4—figure supplement 2). Second, we now also provide a scheme which describes a method inspired by Oster et al., in which phase contrast imaging can be used to determine the Iodixanol concentration required to image the sample at RI-matched conditions (see Figure 4—figure supplement 2). Together, these two additions to the manuscript now accomplish the requested user guide for determining the optimal concentration of Iodixanol.

*Reviewer #2:*

*Review of "A tunable refractive index matching medium for live imaging cells, tissues and model organisms" by Boothe et al.,.*

*This article presents a novel method for improving the resolution and useful penetration depth of fluorescence microscopy of live organisms. The authors define the rationale of the method as a well-known problem in the field of microscopy; namely, the mismatches of refractive indices (RIs) between the optical media through which the emitted fluorescence travels (cytosol, aqueous mounting media, glass coverslips, and objective immersion media) cause spherical aberration in the native images formed. This increased spherical aberration causes decreases in resolution, especially at depths far from the coverslip surface. The method presented in this paper serves as a solution to lower the effective spherical aberration of silicone oil immersion objectives by changing the RI of the media in which the live sample is mounted in order to better match the RI of the objective immersion medium. The authors demonstrate the usability of this RI-matching compound, Iodixanol, with a variety of live model organisms. The authors also demonstrate the increase in fluorescence intensity from live fluorescent organisms that are mounted in a RI-matched media, as compared with the normal aqueous media.*

*This article proposes a fantastic method for increasing the quality of fluorescent images of live organisms. Microscopists have been able to match RIs of fixed samples for years to increase image quality, but this is the first method detailing the possibility of RI matching in live samples with no observable effects on sample vitality. The authors comprehensively demonstrate the non-toxicity of Iodixanol on three common model organisms with varying degrees of complexity. The authors also effectively show that dilution of Iodixanol into traditional mounting media can easily tune the RI of the solution in a linear fashion, which will be useful to microscopists to easily implement with their own samples. The authors are very quantitative in their analysis of the increase in fluorescence intensity in a biological sample as a result of their RI matching method (Figure C,D).*

Thanks for the positive assessment of our work.

*However, the authors fail to quantitatively support their assertion that their RI-matching method increases the practically achievable spatial resolution (in x/y or in z) of the system. There is no measurement or mention of the point-spread function of the objective, which is the true physical definition of the resolution of a microscope. Despite quantitatively showing that the intensity of fluorescence is higher at greater depths, the authors have not shown that the full-width at half maximum intensity (FWHM) of the objective's PSF remains more consistent with their RI-matching method than with traditional mounting methods. Without these data, the claims of providing "substantial improvements in achievable imaging depth" are not yet validated, since a deep image with a high signal-to-noise ratio (SNR) but low spatial resolution only ameliorates one of the two main issues that results from RI-mismatched spherical aberration. The data presented are encouraging and exciting as a live-cell microscopist, but ultimately incomplete at this point with respect to the conclusions.*

Absolutely, we fully agree and have added extensive quantitative data on the resolution effects of Iodixanol supplementation (see below).

*Before recommending this paper for publication, I would make the following suggestions for experiments:*

*1) A quantitative comparison of spatial resolution in three dimensions between traditional media and Iodixanol-supplemented media, using an accepted resolution standard. I would suggest sub-diffraction limit multicolor fluorescent beads (100 nm diameter for high NA) embedded in 1-2% agarose. Fluorescent images of the beads should be acquired as three-dimensional stacks (similar parameters to the images presented in Figure 3) in order to quantify the effect that the Iodixanol has on the lateral (x/y) and axial (z) resolution of the objective. Since the beads will be below the diffraction limit, a three-dimensional gaussian intensity profile (PSF) can be fit to beads that are suspended at various depths away from the coverslip. In order to maximize the relevance of this quantification to most live-cell microscopists, I would recommend quantifying the PSFs using green (~500 nm) and red (~600 nm) fluorescence emission, as these wavelengths correspond to the two most common classes of biological fluorophores used in live-cell fluorescence microscopy (GFP and RFP). Additionally, I would recommend varying the concentration of Iodixanol in these experiments, as the refractive index matching will most likely correct for spherical aberrations at a subset of focal depths, rather than at every depth from the surface of the coverslip to the maximum working distance of the objective. I would recommend reporting the depth of the focal plane about which the spatial resolution is best for several useful concentrations of Iodixanol.*

Done and thanks for these insightful suggestions. We have now quantified resolution in vitro and in vivo and the added data demonstrate quantitatively that RI tuning indeed enhances the spatial resolution.

Firstly, as suggested by the reviewer we now compare in our in vitro experiments sub diffraction sized beads mounted in aqueous agarose (RI = 1.333) with beads mounted in RI-matched agarose tuned to the specific RI of the silicon immersion oil used for imaging (RI = 1.401). RI matching leads to a narrower PSF with a significantly higher intensity and an improved resolution in both lateral (328 ± 15 vs 254 ± 5 nm) and axial (886 ± 26 vs 576 ± 10 nm) directions at 488nm excitation. These data are now displayed in Figure 1 and [Supplementary-material SD5-data] along with results of improved PSFs by Iodixanol supplementation at 638 nm.

Secondly, we injected sub diffraction sized beads into live zebrafish embryos and demonstrate in vivo that lateral resolution can be significantly improved by Iodixanol supplementation especially at larger distances from the coverslip (918 ± 50 vs 792 ± 28 nm at 150µm depth, 638nm excitation). These data are now shown in Figure 4 and [Supplementary-material SD5-data]. Unfortunately, due to the fast dithering of the small beads in live samples it was impossible to record a proper PSF in axial direction. But as our in vitro data suggests, with improvement of lateral resolution an improvement of axial resolution can also be expected (Figure 1) and is qualitatively demonstrated in Figure 3 and Figure 3—figure supplement 1.

*2) Perform a test of common drug solubility/activity in Iodixanol. While I am convinced of the non-toxicity of Iodixanol on vitality and growth of living organisms, many microscopists introduce inhibitory drugs into the mounting media to observe the effects that the drugs have on the organism at a sub-cellular level. Many of these microscopists would most likely not use Iodixanol in mounting media if Iodixanol inhibited the solubility or activity of their experimental drugs. I would suggest a commonly used DMSO-soluble drug, such as nocodazole, which has well-characterized organismal and sub-cellular phenotypes in many model organisms, and observe the differences (if any) of the drug's effect on the organisms' phenotype compared to traditional mounting media supplemented with the drug.*

We treated HeLa cells with Nocadozole with and without Iodixanol and conclude that microtubules depolymerize even in presence of 30% Iodixanol. We did not include these data in the manuscript, as we do not want to generally rule out effects on drug efficacies on basis of n =1 and the compound screen that would be required to support this statement is clearly beyond the scope of this manuscript. On the other hand, we believe that the mention of normal agarose polymerization, growth, embryonic development and regeneration all indicate that Iodixanol does not greatly change molecular interactions.

Author response image 1.HeLa cells stably expressing H2B-mCherry were exposed to 300 nM Nocodazole for 2.5 hrs.Nocodazole treatment leads to depolimerization of microtubules stained with 0.2 nM SiR-Tubilin in both complete media and media supplemented with 30% Iodixanol. Scale bar = 10 µm. Images of live HeLa cells were acquired with a spinning disc confocal microscope.**DOI:**
http://dx.doi.org/10.7554/eLife.27240.017

*3) Measure and report the following physical parameters of Iodixanol at various concentrations: viscosity, autofluorescence, dispersive properties, the pH buffering capacity, and the refractive index at various wavelengths.*

1) We have included the following quantifications into the manuscript:

The viscosity of Iodixanol has been reported by commercial distributors (see https://goo.gl/I4owRU) and this reference is reported in the methods section.

2) We measured and now report that 60% Iodixanol stock solution has no significant autofluorescence at the commonly used excitation wavelengths of 405, 488, 560 and 640 nm. As a negative control we used PBS and as a positive control fluorescent bead suspensions. These data are now displayed in Figure 1 and Figure 1—figure supplement 1.

3) The dispersive properties out of which the surface tension is for biological applications the most relevant parameter has been reported for Iodixanol (89.6 mN/m; https://goo.gl/VsCxbJ) and is similar to that of water (72.0 mN/m). One should consider that the surface tension of Iodixanol is concentration dependent as previously reported (DOI: 10.1183/09031936.02.00257902). We do not have the laboratory equipment to measure surface tension at various Iodixanol concentrations in biological media and refer to the mentioned sources for further details. Collectively, we did not find the dispersive properties of Iodixanol to be disadvantageous in our experiments.

4) We performed a pH-titration of Iodixanol and compared its pH buffering capacities to water and PBS. We conclude from these experiments that Iodixanol has no strong pH buffering capacity within physiological relevant pH ranges and behaves similar to the pH buffering capacity of water. These data are now displayed in Figure 1.

5) Unfortunately, we do not have the technical equipment to measure the RI in dependence of the excitation wavelength. The equipment used for RI measurements measured the RI at a wavelength of 589.3 nm. We now added this important parameter to the methods section (subsection “Determination of Iodixanol’s physical prop”).

*4) Use a higher-resolution, oil-immersion lens (numerical aperture {greater than or equal to} 1.4) with high refractive index immersion oil (n ~= 1.5). The authors mainly focus on silicone-immersion objectives, which offer greater working distance but lower native resolution and brightness than high-NA objectives. Microscopists that use these objectives also experience spherical aberration; in fact, the spherical aberration more sharply affects the resolution as a function of depth. I am interested to know if this RI-matching method improves the resolution and brightness of images using high-NA objectives at depths under 10μm from the coverslip.*

Done and this is a valid point. However, we want to highlight that the highest possible RI Iodixanol can compensate for is RI 1.43. As correctly stated by the reviewer, common oil immersion media display an RI > 1.5. Iodixanol can therefore not fully compensate for spherical aberrations caused by RI mismatches between immersion oil and culture media. We performed the requested experiment at the highest Iodixanol concentration regardless using a 100x NA 1.40 objective with oil immersion (RI = 1.518). As expected the lateral resolution does only slightly improve (294 ± 10 nm vs 258 ± 7 nm) whereas no significant changes were detected in the axial resolution (696 ± 25 vs 694 ± 17 nm) at 488nm excitation. Since this is obviously a result of an insufficient RI match, we did not include the data from Figure 6 in the manuscript.

Author response image 2.**DOI:**
http://dx.doi.org/10.7554/eLife.27240.018

We conclude from these data that Iodixanol is indeed capable of improving lateral resolution at high NA Oil immersion objectives.

Finally, we also want to highlight that these data do not exclude that Iodixanol can provide a significant improvement with high NA lenses independent of the used immersion media. When additional spherical aberrations are introduced by the sample itself due to RI mismatches between sample and media we expect Iodixanol to have a major improvement on image resolution (as shown in Figure 4). Importantly, we see Iodixanol in the first place as an RI matching agent to overcome RI mismatches between culture media and sample and not between culture media and objective immersion liquid, which we believe is clearly stated in the text.

Overall, we again thank our reviewers for their insightful and constructive comments. We believe that we have addressed them all, resulting in a manuscript that, in our opinion, now provides a useful resource for the life imaging research community.